# Caregiver Perceptions of Child Diet Quality: What Influenced Their Judgment

**DOI:** 10.3390/nu14010125

**Published:** 2021-12-28

**Authors:** Lijing Shao, Yan Ren, Yanming Li, Mei Yang, Bing Xiang, Liping Hao, Xuefeng Yang, Jing Zeng

**Affiliations:** 1School of Public Health, Wuhan University of Science and Technology, Wuhan 430065, China; 202009703046@wust.edu.cn (L.S.); 202009703044@wust.edu.cn (Y.R.); liyanming@wust.edu.cn (Y.L.); 2School Research Center for Woman and Child Health, Wuhan University of Science and Technology, Wuhan 430065, China; yangmei88@wust.edu.cn (M.Y.); Xiangbing@wust.edu.cn (B.X.); 3Department of Nutrition and Food Hygiene, Tongji Medical College, Huazhong University of Science and Technology, Wuhan 430065, China; haolp@mails.tjmu.edu.cn (L.H.); xxyxf@mails.tjmu.edu.cn (X.Y.)

**Keywords:** child, diet quality, caregiver perception, CCDI

## Abstract

This study aimed at assessing the correctness of a caregiver’s perception of their child’s diet status and to determine the factors which may influence their judgment. 815 child-caregiver pairs were recruited from two primary schools. 3-day 24-h recall was used to evaluate children’s dietary intake, Chinese Children Dietary Index (CCDI) was used to evaluate the dietary quality. Multivariate logistic regression models were used to explore the factors that could influence the correctness of caregiver’s perception. In the current study, 371 (62.1%) children with “high diet quality” and 35 (16.1%) children with “poor diet quality” were correctly perceived by their caregivers. Children who were correctly perceived as having “poor diet quality” consumed less fruits and more snacks and beverages than those who were not correctly perceived (*p* < 0.05). Obese children were more likely to be correctly identified as having “poor diet quality” (OR = 3.532, *p* = 0.040), and less likely to be perceived as having “high diet quality”, even when they had a balanced diet (OR = 0.318, *p* = 0.020). Caregivers with a high level of education were more likely to correctly perceive children’s diet quality (OR = 3.532, *p* = 0.042). Caregivers in this study were shown to lack the ability to correctly identify their children’s diet quality, especially amongst children with a “poor diet quality”. Obesity, significantly low consumption of fruits or high consumption of snacks can raise caregivers’ awareness of “poor diet quality”.

## 1. Introduction

Childhood obesity is a major health problem both in the developed and developing countries [1]. In 2015, amongst the 20 most populous countries, China ranked first in terms of the number of obese children [2]. The national prevalence estimates of obesity and overweight among Chinese children were increased to 10.4% for children younger than 6 years and 19% for children aged 6 to 17 years in 2019 [3], and this trend will only continue to increase with the continuous development of the economy. 

Diet quality plays an important role in weight control [4,5]. Diets containing adequate fruits and vegetables and less energy-dense nutrient-poor (EDNP) foods can not only help people control their weight [6,7,8], but also reduce the risk of negative health outcomes and all-cause mortality [9,10,11,12,13]. Previous studies have revealed that more than 60% of Chinese school-age children failed to meet the recommendations issued by Chinese Dietary Guidelines regarding fruit and vegetable intake [14,15,16], which undoubtedly will have a detrimental impact on their development. Given that dietary habits developed in childhood can be carried into adulthood and impact long-term health outcomes [17,18], early detection of poor diet quality in children and taking timely action is of great importance.

In order to cultivate healthy eating habits at an early stage, researchers and governments have started to develop nutrition education activities targeting school-age children. However, children (especially young children) have little autonomy over their food choices, with almost all foods being provided by their caregivers. Thus, attitudes of caregivers are crucial to the effectiveness of these nutrition interventions [19,20]. If they are unable to recognize unhealthy habits and help their child to change these, the effect of health education will be minimal. Previous research found parents with correct perception of their child’s overweight status were more likely to make changes to their children’s lifestyles and participate in healthy lifestyle behaviors with their children [21,22]. However, it also found that those who failed to recognize their child’s weight status were less motivated to address the problem [21,22]. Thus, we have reason to believe that caregivers with correct perception of their children’s poor diet quality could also be more willing to participate in nutrition promotion activities with their children, which could greatly improve the effectiveness of health interventions.

However, limited information is available concerning the caregiver’s perception of their child’s diet quality, with previous studies focusing mainly on caregiver’s perception of their child’s weight status. It appears common for caregivers to underestimate their child’s weight; a meta-analysis indicated that nearly half of parents underestimated their children’s overweight/obese status and a significant minority underestimated their children’s normal weight status [21]. Furthermore, Reyes et al. found that 50% of caregivers of children aged 2–18 years underestimated their children’s weight status [23]. Boys’ weight status was more likely to be underestimated compared to girls [24], potentially due to different perceptions of an ideal body shape for boys and girls. Girls were more likely to be correctly perceived as overweight, whilst overweight boys however were more likely to be regarded as strong rather than overweight. 

Caregiver’s correct perception of children’s health issues is crucial to the maintenance of children’s health. It has already been reported that caregiver’s perception of children’s weight status was sub optimal, potentially indicating poor health literacy among caregivers, therefore it could also be assumed that their perception of children’s diet quality could require improvement. However, there is still a lack of research regarding the perception of caregivers of children’s diet quality. Therefore, in the current study, we aimed to examine the correctness of caregiver’s perception of their child’s diet quality and to investigate the factors which may influence their judgment.

## 2. Materials and Methods

### 2.1. Study Population and Ethical Statement

Participants were recruited from two primary schools using cluster sampling in Hongshan district, Wuhan, China in April 2016. Considering that the children’s diet needed to be self-reported and the cognitive ability of students in lower grades is limited, in the current study we selected students in grades 3rd to 6th and their caregivers as subjects. Initially, 1132 eligible child-caregiver pairs were recruited. Of these, 317 were excluded due to incomplete information or not providing consent. Therefore, this analysis was based on a final sample of 815 child–caregiver pairs. Figure 1 provides an overview of the recruitment process. This study was conducted according to the guidelines of the Declaration of Helsinki and all procedures involving human subjects were approved by the Ethics Committee of Wuhan University of Science and Technology (No. 201519). All parents gave written informed consent and children gave assent. 

### 2.2. Demographic Characteristics

Data relating to socio-demographic status and caregiver’s perception were collected using customised, self-rated questionnaires. Information on age, gender, frequency and location of meals were collected for children, whilst information on age, gender, relationship to the children, annual household income and level of education was collected for caregivers. Members of the research team including faculty and postgraduates of the school of public health helped students fill out the questionnaire in the school setting. Caregivers’ questionnaires were brought home by the students; the primary caregivers of the children were then asked to fill in the questionnaire, and after filling it in, the questionnaires were brought back to school by the students the following days.

### 2.3. Diet Survey and Evaluation

3-day 24-h recall was used to evaluate children’s dietary intake. The children were asked to recall all food and beverages consumed in the past 24 h for three consecutive days, the investigators went into the classroom after lunch, and using food size reference models recorded all the food consumed by each student in the past 24 h (including drinks, snacks, inter-class meals, etc.). Data relating to eating behaviors were collected through interviews. Children’s dietary reference values issued by Chinese Dietary Guidelines vary by age and sex [25], so food intakes were converted to food density (g/1000 kcal) for the purpose of comparison (except for snacks and beverages). The daily dietary intake of calories and nutrients were calculated using the China Food Composition 2004 and were presented as an average intake over the 3-day period.

The Chinese Children Dietary Index (CCDI) was used to evaluate the diet quality of the children involved in the study based on the Chinese dietary intake recommendations (Chinese Dietary Guidelines and Chinese Dietary Reference Intakes) and health-promoting behaviors. It was developed by Guo Cheng et al., though it is not widely used at present, the validity of CCDI in evaluating the dietary status of Chinese school-age children has been verified in several studies [14,26,27].

The CCDI contains 16 items in four sections. The highest score for each item is 10 points, therefore the CCDI has a total score of 160 points, with higher scores representing better diet quality. The scoring scheme is based on the amounts and types of nutrients or foods that the children consumed, and whether they exhibited health-promoting behaviors. The first part evaluates the intake of 8 food groups (grains, vegetables, fruits, dairy and dairy products, soybean and its products, fish and shrimp, meat and eggs), water and sugar-sweetened beverages. The second part evaluates intakes of vitamin A, fatty acids and dietary fiber. The third part evaluates food diversity and the final part evaluates dietary behaviors, including breakfast, dinner and energy intake. Criteria relating to maximum and minimum score of food and nutrients were derived from recent age- and sex-specific dietary reference values [25,28], and the criteria for food diversity and diet behaviors were derived from the Chinese Dietary Guidelines. Details relating to these have been published previously [14].

Since sedentary behaviors or water intake were not assessed in this study, the CCDI was slightly modified (Table 1). The item “drinking water” was removed, and the score of “energy balance” was determined only by total energy intake, rather than total energy intake and sedentary behaviors. “Fatty acid” was changed to total fat intake, and the cut-offs were derived from the Chinese Dietary Guidelines. Following modification, scores ranged from 0 to 150. Scores above 60% were considered acceptable, therefore a score of over 90 points was considered “high diet quality” and a score below 90 points was considered “poor diet quality”.

### 2.4. Caregiver Perception

Caregiver’s perception of their child’s diet was assessed using the question ‘How would you describe your child’s diet quality?’ and they were given three choices: high diet quality, poor diet quality, and unknown (reason is needed). The accuracy of this perception was assessed by comparing the caregiver’s perception with the child’s actual diet quality. Caregivers who had a different perception of their child’s diet quality compared to what their child’s diet quality actually were deemed to have incorrectly perceived their child’s diet quality.

### 2.5. Covariates

In this study, the primary caregiver was defined as the person who takes care of the child or prepares food for the child most often. The caregiver’s level of education was defined as the highest degree that the primary caregiver had completed at the time of the survey. Family income refers to the average annual household income, including but not limited to wages, self-employed income and agricultural income.

Data relating to the children’s weight and height were obtained from a physical examination made by Wuhan ChangeDong Hospital; height was measured with a precision of 0.1 cm and weight was measured with a precision of 0.1 kg. This information was then used to calculate Body Mass Index (BMI) and obesity was defined according to the BMI cutoffs points issued by the National Health Commission of the People’s Republic of China [29].

### 2.6. Statistical Analysis

Descriptive statistics included frequency and percentages for categorical variables and median (P25, P75) for continuous variables that were not normally distributed. Rank sum test for continuous variables with a non-normal distribution and chi-square tests for categorical variables were used. Variables found to be statistically significant in univariate analysis and variables that are thought to be associated with diet quality (such as gender, family income, et al.) were included in multivariate logistic regression, to explore the factors that could influence the correctness of caregiver’s perception. Statistical analyses for this study were performed using Stata (version 13.0; StataCorp, College Station, TX, USA). Differences were considered significant if *p* < 0.05. 

## 3. Results

### 3.1. Participant Characteristics

Demographic characteristics of children based on their diet quality can be seen in Table 2. More than half of the children were migrants (57.9%). Most caregivers were not well educated and had only completed middle school (51.8%). Half of the families had an average annual income lower than 50,000 CNY (50.7%). 26.7% of the children were classed as having a poor-quality diet, with boys and overweight/obese children being more likely to have a poor-quality diet (*p* < 0.05) (Table 2).

### 3.2. Caregiver Perception

Among the 597 children with “high diet quality”, 371 (62.1%) were correctly perceived by their caregivers, whilst among the 218 children with “poor diet quality”, just 35 (16.1%) were correctly perceived (Table 2). Whether in the “high diet quality” group or the “low diet quality” group, most parents believe that their children had a high diet quality (*p* > 0.05). 

Of the 815 caregivers, 209 (25.6%) caregivers could not make a clear judgment on their child’s diet. These caregivers did not exhibit any significant differences in relation to educational level, family income and their child’s gender, weight status and diet scores when compared to caregivers with clear judgment (*p* > 0.05). Therefore, in further analysis only those child-caregiver pairs in which the caregiver had given a clear judgment were included (*n* = 606) (Table 3).

### 3.3. Children’s Daily Food Intake

Table 4 shows that children with “poor diet quality” had lower intakes of vegetables, fruits, fish, eggs, beans and milk (*p* < 0.01), and higher intakes of grains, snacks and beverages (*p* < 0.01). Children who were correctly perceived by their caregivers as having “poor diet quality” consumed less fruits than those who were not correctly perceived, and they were more likely to consume snacks and beverages than the other groups (*p* < 0.05).

### 3.4. Influence Factors of Caregiver’s Perception of Diet

Migration, child’s gender, caregiver’s association with the child, family income and breakfast habits did not affect the correctness of the caregiver’s perception of the child having poor diet quality. Caregivers were more likely to identify “poor diet quality” among children who consumed less fruits (OR = 0.989, *p* = 0.031) or those eat more snacks (OR = 1.074, *p* = 0.004). Obese children were more likely to be correctly identified as having “poor diet quality” (OR = 3.532, *p* = 0.040), and these children were also more likely to be perceived as having “poor diet quality” even when they had a balanced diet (OR = 0.318, *p* = 0.020). Moreover, caregivers with a high level of education were more likely to correctly perceive children’s “high diet quality” (OR = 3.532, *p* = 0.042) (Table 5).

## 4. Discussion

The aim of the current study was to examine the correctness of caregiver’s perception of their child’s diet quality and to investigate the potential factors that may influence this. To our knowledge, this is one of the first studies to pair caregivers with their child to examine caregiver’s perception of child diet quality, given that previous studies have focused on caregiver’s perception of their child’s weight status [21,22,23,24]. In the present study, our results showed that it is common for caregivers to be unable to identify the diet quality of their child. This was particularly noticeable in children with “poor diet quality”, with less than a fifth of children with a poor-quality diet being correctly perceived by their caregivers. This is worrying to note; if caregivers are not aware of their child’s poor diet quality, they might be less likely to change their dietary behaviors, and this could lead to an increased risk of obesity, malnutrition or chronic diseases in the future. 

Children in the “poor diet quality” group had lower intakes of several kinds of foods apart from grains and meat, and their snacks and beverages intakes were significantly higher. However, it was only extremely low intakes of fruits or high intakes of snacks that caregivers seemed to be aware of, they did not take other food types (including vegetables, fish, beans, eggs, and milk) into consideration. According to our survey, children with “poor diet quality” had significantly lower intakes of fish, eggs, beans and milk, with nearly half of them having never consumed these foods over a 3-day period, which may suggest that these kinds of foods were not frequently served as part of school lunches. A previous study found that the supply of milk, beans, fish and eggs at school lunches in Shanghai, China, did not reach the recommended level [30]. Although Shanghai is a first-tier city in China with high standard of living, the schools investigated in this study were average public primary schools in Wuhan, which are less likely to provide such foods. These foods are therefore more likely to be prepared by caregivers at breakfast or dinner time. The lower intake of these foods may be due to the caregivers not understanding the benefits of them, or not regularly preparing these foods for their children. Therefore, they did not take these foods into consideration when judging their children’s diet quality. Most of the children in this study were eating a certain quantity of vegetables, however in those with “poor diet quality”, their consumption was significantly lower. A potential reason why caregivers could not make accurate judgments on this insufficient intake might because they were not aware of the number of vegetables that children should be eating. Therefore, they would not be able to know whether their children’s consumption matched the recommended number while making the judgment.

Generally, highly educated caregivers have more access to correct health information [22,31] and are more likely to make accurate judgments about their children’s diet. This was highlighted in this study; caregivers with a college degree or above were able to make more accurate judgments relating to children with “high diet quality”, but not “poor diet quality”. Caregivers play an integral role in shaping children’s eating behavior, and their attitude towards diet can influence children’s eating habits [32]. Compared to the caregivers with children in the “high diet quality” group, those with children in the “poor diet quality” group might be less concerned about diet quality and therefore were not as aware of their children’s diet habits, which may have led to their children having a poor-quality diet. Therefore, even if they did have adequate nutrition knowledge, their lack of concern and understanding of children’s diet could prevent them from making accurate judgments.

Obese children with both “high diet quality” or “low diet quality” were more likely to be perceived as having a “poor diet quality”, even when their diet quality was in fact adequate. Caregivers were aware that improper diet can lead to obesity, however they may have ignored the influence of genetics, sports, mental health and other factors [33], which indicates that caregivers may not have comprehensive health knowledge.

Regardless of level of education and family income, the inability to correctly identify poor diet quality appears to be a common phenomenon. It has been suggested previously that the action of parents’ judging their child’s actual weight is not a cognitive task, but an emotional evaluation [34]. This may also be the case for diet judgment; caregivers tend to believe that their children have a “high quality diet” when they are unsure whether their children’s diet is up to the standard or not. This tendency may also lead caregivers to be less motivated when participating in diet promotion courses [21,22]. Thus, it is necessary for them to be taught how to judge diet accurately.

Therefore, in a future nutrition intervention program, caregivers should firstly be educated about the importance of a healthy diet for children, which may improve their awareness of their child’s diet. There is the potential to develop tailored nutrition education for caregivers, so that they can have adequate knowledge relating to nutrition, be aware of the standards of a healthy diet for children and learn cooking skills, so that they can spot any deficiencies in children’s diet and correct them timely. Finally, caregivers should be informed that causes of obesity are multifactorial, such as dietary, exercise and genetics et al. They should understand that diet is not the only factor that can cause obesity, and similarly, obesity does not necessarily mean the child has an unhealthy diet.

### Strengths and Limitations

There are several limitations in this study. Firstly, as the participants were recruited from two local schools, the sample is not nationally representative and therefore the results of this study cannot be generalized to other samples. Secondly, in order to improve participant compliance, the 3-day dietary recalls were performed during school hours, therefore the data may not represent children’s diet on the weekends. At the same time, best practice for 24-h recalls is three unannounced days, not three consecutive days, so this may lead to bias caused by changes of food behaviors. Thirdly, since there are no widely accepted indices to measure diet quality in Chinese children, CCDI was used to assess children’s diet quality. Although not widely used, the effectiveness of CCDI has been proved in another study in children of the same age [14]. Finally, although caregivers’ perceptions of children’s diet quality were obtained by asking only one question, which may not be comprehensive, similar one question-based methods have been used previously in research relating to caregivers’ perception of children’s weight status [23,24]. Despite these limitations, this study is one of the first to examine caregiver’s perception of child diet quality and the factors that can influence this, and these results have highlighted the inadequate health awareness among parents or other caregivers and can provide direction for future health interventions.

## 5. Conclusions

Caregivers in this study were shown to lack the ability to correctly identify their children’s diet quality, especially amongst children with a “poor diet quality”. Obesity, significantly low consumption of fruits or high consumption of snacks can raise caregivers’ awareness of “poor diet quality”. However, if the child was not classed as obese or they did not show any special rejection/preference for fruits and snacks, their poor diet quality was less likely to be noticed. Caregivers’ judgment was also influenced by children’s consumption of certain types of food and their body type, rather than the child’s overall diet. Such information may be valuable for the prevention of obesity and malnutrition among children through improving caregivers’ awareness of child diet quality.

## Figures and Tables

**Figure 1 nutrients-14-00125-f001:**
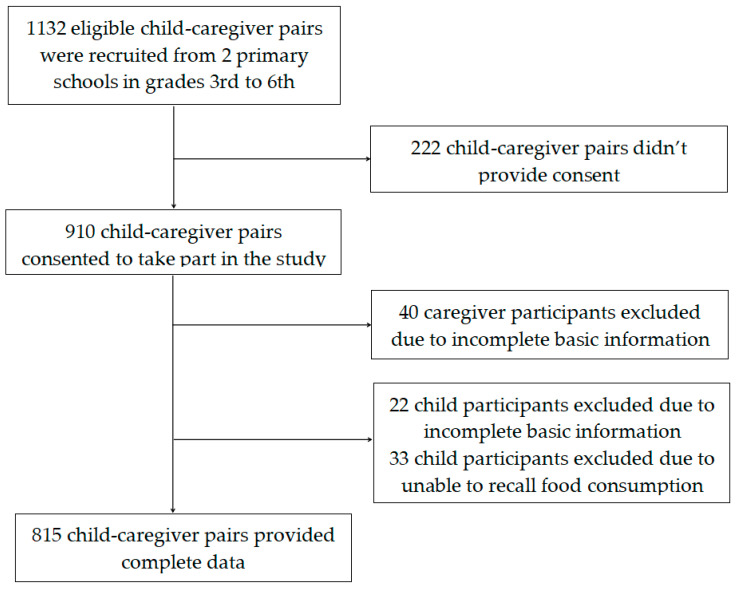
Flow diagram of subject recruitment.

**Table 1 nutrients-14-00125-t001:** Adjusted scoring system for the components of the CCDI, a measure of overall diet quality for Chinese school-aged children.

CCDI Component	Range of Score (Points)	Criteria for Maximum Score	Criteria for Minimum Score
Food Groups ^a^			
Grains ^b^	0–10	140–160 g/1000 kcal	0 or >320 g/1000 kcal
Vegetables ^c^	0–10	≥175 g/1000 kcal	0 g/1000 kcal
Fruits	0–10	≥110 g/1000 kcal	0 g/1000 kcal
Dairy and dairy products	0–10	≥110 g/1000 kcal	0 g/1000 kcal
Soybeans and its products	0–10	≥17 g/1000 kcal	0 g/1000 kcal
Meat	0–10	25–35 g/1000 kcal	0 or >70 g/1000 kcal
Fish and shrimp	0–10	≥30 g/1000 kcal	0 g/1000 kcal
Eggs	0–10	12.5–22.5 g/1000 kcal	0 or 45 g/1000 kcal
SSBs ^d^	0–10	0 mL/day	≥1 serving/day
Nutrients			
Vitamin A ^e^	0–10	≥100% RNI/day	0% RNI/day
Fat ^f^	0–10	20–30% E/day	0% or >60% E/day
Dietary fiber	0–10	≥14 g/1000 kcal	0 g/1000 kcal
Diet variety	0–10	>1 serving of food from each of these groups (grains, vegetables, fruits, dairy/beans, and meat/fish/eggs)	<1 serving of food from each of these groups (grains, vegetables, fruits, dairy/beans, and meat/fish/eggs)
Behaviors			
Breakfast and dinner	0–10	Eating breakfast and having dinner with parents regularly	Skipping breakfast and not having dinner with parents regularly
Energy balance	0–10	0.9 EER ≤ EI ≤ 1.1 EER	EI = 0 or EI ≥ 2.2 EER
CCDI total score	0–150		

Abbreviations: CCDI, Chinese Children Dietary Index; SSBs, sugar-sweetened beverages; RNI, recommended nutrient intakes; E, energy; EER, estimated energy requirement; EI, energy intake. ^a^ To characterize diet quality, consumption of food groups were expressed on a per −1000-calorie basis in the CCDI. ^b^ Given that grains, meat, and eggs should be consumed moderately, consumption between the lowest and highest recommended amount per 1000 kcal according to the Chinese Dietary Guidelines (2007) was chosen as the standard for the maximum score. ^c^ Vegetables, fruits, dairy and dairy products, soybeans and its products, and fish and shrimp should be consumed sufficiently. The lowest recommended amount per 1000 kcal according to the Chinese Dietary Guidelines (2007) was chosen as the standard for the maximum score for these food groups. ^d^ SSBs were defined as beverages with added sugar, such as lemonades, fruit drinks, ice teas, soft drinks (soda pop), sport drinks, tea and coffee drinks, and sweetened milks. One serving is 250 mL. ^e^ RNI of vitamin A: 500 μgRAE/day (children aged 7 to 10 years), 630 μgRAE/day (girls aged 11 to 12 years), 670 μgRAE/day (boys aged 11 to 12 years). ^f^ Consumption of fat within the AMDR (Acceptable Macronutrient Distribution Range) was chosen as the standard for the maximum score.

**Table 2 nutrients-14-00125-t002:** Characteristics of study participants ^a^.

Characteristic		Child Diet Status		χ^2^	*p*-Value
		High Diet Quality (*n* = 597)	Poor Diet Quality (*n* = 218)		
Gender	Boys	243 (40.7%)	114 (52.3%)	8.714	0.003
	Girls	354 (59.3%)	104 (47.7%)		
Primary caregiver	Mother	400 (67.0%)	151 (69.3%)	1.664	0.645
	Father	102 (17.1%)	37 (17.0%)		
	Grandparents	63 (10.5%)	23 (10.5%)		
	Others	32 (5.4%)	7 (3.2%)		
Family income	<50,000¥	298 (49.9%)	115 (52.7%)	0.514	0.473
	≥50,000¥	299 (50.1%)	103 (47.3%)		
Caregiver’s educational level ^b^	Primary school	44 (7.4%)	21 (9.6%)	3.646	0.302
	Middle school	258 (43.2%)	99 (45.4%)		
	High school	231 (38.7%)	83 (38.1%)		
	College	64 (10.7%)	15 (6.9%)		
Weight status	Normal weight	493 (82.6%)	164 (75.2%)	6.496	0.039
	Overweight	73 (12.2%)	34 (15.6%)		
	Obesity	31 (5.2%)	20 (9.2%)		
Caregiver’s perception	High diet quality	371 (62.1%)	131 (60.1%)	3.012	0.222
	Poor diet quality	69 (11.6%)	35 (16.1%)		
	Unknown	157 (26.3%)	52 (23.8%)		

^a^ Data are presented as counts (percentages). ^b^ Caregiver’s educational level: represented as the highest degree of the primary caregiver.

**Table 3 nutrients-14-00125-t003:** The correctness of caregiver’s perception of diet status of children in with differing diet quality.

Characteristic		High Diet Quality ^a^ (*n* = 440)			Poor Diet Quality ^a^ (*n* = 166)		
		Total	Correct ^b^	*p*	Total	Correct ^b^	*p*
Gender	Boys	170	142 (83.5%)	0.718	95	20 (21.0%)	0.991
	Girls	270	229 (84.8%)		71	15 (21.1%)	
Primary caregiver	Mother	303	250 (82.5%)	0.178	114	24 (21.0%)	0.889
	Father	67	57 (85.1%)		27	5 (18.5%)	
	Grandparents/Others	70	64 (91.4%)		25	6 (24.0%)	
Family income	<50,000¥	214	173 (80.8%)	0.051	83	16 (19.3%)	0.568
	≥50,000¥	226	198 (87.6%)		83	19 (22.9%)	
Caregiver’s educational level ^c^	Primary school	34	25 (73.5%)	0.096	18	4 (22.2%)	0.923
	Middle/High school	352	297 (84.4%)		136	29 (21.3%)	
	College	54	49 (90.7%)		12	2 (16.7%)	
Weight status	Normal weight	365	312 (85.5%)	0.069	121	22 (18.2%)	0.012
	Overweight	54	45 (83.3%)		29	5 (17.2%)	
	Obesity	21	14 (66.7%)		16	8 (50.0%)	

^a^ High diet quality (or poor diet quality) represents the child’s diet status which assessed by CCDI. ^b^ Correct means the caregiver’s recognition is consistent with the child’s diet status which assessed by CCDI. ^c^ Caregiver’s educational level represented as the highest degree of the primary caregiver.

**Table 4 nutrients-14-00125-t004:** Food consumed by each diet quality group according to diet status and caregiver’s perception (M (P25, P75)).

Food Group	High Diet Quality ^a^				Poor Diet Quality ^a^			
	Correct ^b^ (*n* = 371)	Incorrect ^b^ (*n* = 69)	*Z*	*p*	Correct ^b^ (*n* = 35)	Incorrect ^b^ (*n* = 131)	*Z*	*p*
Grain (g/1000 kcal)	158.0 (145.0, 173.3)	157.5 (140.6, 172.4)	0.783	0.434	170.5 (160.4, 185.5)	167.5 (154.8, 178.3)	1.045	0.296
Vegetables (g/1000 kcal)	102.8 (85.0, 122.7)	103.1 (85.0, 125.4)	0.123	0.902	95.6 (77.4, 106.6)	93.6 (81.0, 107.3)	−0.107	0.915
Fruit (g/1000 kcal)	106.8 (68.3, 150.7)	117.1 (87.2, 145.6)	−0.902	0.367	20.2 (0.0, 61.4)	48.3 (0.0, 93.2)	−2.380	0.017
Meat (g/1000 kcal)	25.8 (18.5, 35.4)	26.8 (19.1, 33.0)	0.317	0.752	24.8 (15.2, 42.6)	24.3 (14.9, 36.1)	0.424	0.672
Fish (g/1000 kcal)	4.9 (0.0, 13.6)	6.3 (0.0, 16.7)	−1.312	0.190	0.0 (0.0, 8.7)	2.8 (0.0, 8.9)	−0.431	0.667
Egg (g/1000 kcal)	11.6 (2.0, 20.4)	13.6 (5.0, 20.2)	−0.260	0.795	0.0 (0.0, 9.9)	0.0 (0.0, 9.6)	0.992	0.321
Beans (g/1000 kcal)	4.4 (0.0, 10.3)	3.1 (0.0, 9.9)	0.630	0.529	0.0 (0.0, 6.2)	0.0 (0.0, 5.0)	0.353	0.724
Milk (g/1000 kcal)	41.2 (0.0, 97.1)	41.7 (0.0, 109.9)	−0.496	0.620	0.0 (0.0, 31.7)	0.0 (0.0, 35.0)	−0.220	0.826
Snacks (g/day)	18.5 (12.8, 24.0)	20.4 (16.7, 26.9)	−1.765	0.078	30.5 (19.0, 38.5)	19.7 (13.5, 28.2)	3.841	<0.001
Beverages (g/day)	34.4 (0.0, 59.6)	30.8 (0.0, 69.4)	0.312	0.755	71.8 (0.0, 160.7)	46.0 (0.0, 76.4)	2.052	0.040

^a^ High diet quality/poor diet quality represents child diet status. ^b^ Correct/incorrect represents whether the caregiver’s recognition is consistent with the child’s diet status.

**Table 5 nutrients-14-00125-t005:** Regression model of the relationship between caregiver’s perception and other socio-demographic predictors, and intake of some kinds of food ^a^.

High Diet Quality Group			
Characteristic		OR (95% CI)	*p*-Value
Family income	<50,000¥	Reference	
	≥50,000¥	1.555 (0.914, 2.645)	0.103
Caregiver’s educational level ^b^	Primary school	Reference	
	Middle/High school	1.932 (0.844, 4.427)	0.119
	College	3.532 (1.046, 11.925)	0.042
Weight status	Normal weight	Reference	
	Overweight	0.858 (0.392, 1.877)	0.701
	Obesity	0.318 (0.121, 0.836)	0.020
**Poor Diet Quality Group**			
**Characteristic**		**OR (95% CI)**	***p*-Value**
Weight status	Normal weight	Reference	
	Overweight	1.109 (0.357, 3.443)	0.858
	Obesity	3.532 (1.056, 11.805)	0.040
Fruits intake		0.989 (0.980, 0.999)	0.031
Snacks intake		1.074 (1.023, 1.129)	0.004
Beverages intake		1.001 (0.995, 1.008)	0.640

^a^ According to the dietary status of children, two logistic regression models were established respectively, the dependent variables were the correctness of the caregivers’ judgment. ^b^ Caregiver’s educational level: represented as the highest degree of the primary caregiver.

## Data Availability

The database used in the study was available from the corresponding author upon reasonable request.

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
