# Peer review of "Caregiver Perceptions of Child Diet Quality: What Influenced Their Judgment"

_nutrients, 2021, doi:10.3390/nu14010125_

Round 1
Reviewer 1 Report
This is an interesting manuscript about the correctness of caregiver’s perception of their child’s diet quality and the investigation of the potential factors that may influence this. Its main contribution is that is a novel topic of interest for prevention of future diseases and with still not so much information and that the number of participitants in the study is high.
General concept comments
The main weakness of the study is that the methodology both to measure the perception of caregivers and the questionnaire used to analyze the children's dietary habits are not adequate. I believe that it is not appropriate to use 24-hour questionnaires to quantify the intake of different food groups (vegetables, fruits, cereals...) since what a person eats for three days is not significant, there can be many circumstances that make it not (the season of the year in which they are for example). For this, it would be more appropriate to use the dietary questionnaires on the frequency of food consumption.In addition, the behavior section of the CCDI survey does not seem adequate to me to measure the quality of the children's diet since it asks if children eat breakfast and meals with their parents regularly, but it does not ask what these meals consist of.
Specific comments
There are quite a lot cited references older than 5 years, please review them and try to find more current ones.
Lines 162-164: What are they based on to use these cut-off points when establishing the quality of the diet?
Table 1: This table indicates which is the criteria to obtain the maximum score in each of the items, but it is not clear how the points are distributed in the range 0-10 when the consumption values ​​are intermediate.
Line 222-224: it should be made clear that this result is not significant.
Table 3: Try to put the table on the same sheet
Table 3: The legend of table 3 is not clear enough
Lines 277-282: As I have mentioned previously, I believe that the 24-hour reminder is not the appropriate dietary questionnaire to assess the consumption of different food groups, so it seems wrong to affirm these results. It may be that when conducting those same questionnaires to the same participants at another time, the results were not the same.
Reviewer 2 Report
This was an interesting study to address a stated gap about caregivers’ perceptions of child’s diet quality. The child-caregiver design was compelling. The following suggestions can improve the paper:
L16: CCDI needs to be defined.
L128-129: Three consecutive days? Best practice for 24-hour recalls is 3 unannounced days to avoid bias in food behavior. Please mention this as a limitation.
L128, L142-143. Mentioning the 24 hour recall twice is confusing. Are these two lines referring to the same questionnaire or a different one? Is the CCDI the 24-hour recall tool used to measure actual dietary intake? Or, is it a questionnaire to measure diet quality and health related behaviors? Please clarify the wording.
L180: Is this a validated tool to measure perceived diet quality? Please clarify. (I see this is addressed briefly in line 339; it would be helpful to mention in the methods also).
L214: Please define RMB
L327-329. I recommend breaking up this long sentence and also using more nuanced language to describe the causes of obesity. As the authors state, causes of obesity are multifactorial, and thus the authors should word the discussion so as to not suggest that exercise or genetics cause obesity.
L354-355: This closing statement is not supported by the study. No obesity prevention claims should be made, or claims regarding the effect of improving caregiver’s awareness. Please modify the wording.
